# Video Forgery Detection Using Multiple Cues on Fusion of EfficientNet and Swin Transformer

## Abstract

The rapid development of video processing technology makes it easy for people to forge videos without leaving visual artifacts. The spread of forged videos may lead to moral and legal consequences and pose a potential threat to people's lives and social stability. So it is significant to identify deepfake video information. Although the previous detection methods have achieved high accuracy, the generalization is poor when facing unprecedented data in the real scene. There are three fundamental reasons. The first is that capturing the general clue of artifacts is difficult. The second is that selecting the appropriate model is challenging in specific feature extraction. The third is that exploiting fully and effectively the extracted features is hard. We find that the high-frequency information in the image and the texture in the shallow layer of the model expose the subtle artifacts. The optical flow of the real video has variations while the optical flow of the deepfake video has rarely variations. Furthermore, consecutive frames in the real video have temporal consistency. In this paper, we propose a dual-branch video forgery detection model named ENST, which integrates parallelly and interactively EfficientNet-B5 and Swin Transformer. Specifically, EfficientNet-B5 extracts the artifacts information of high frequency and texture in the shallow layer of the model. Swin Transformer captures the subtle discrepancies between optical flows. To extract more robust face features, we design a new loss function for EfficientNet-B5. In addition, we also introduce the attention mechanism into EfficientNet-B5 to enhance the extracted features. We conduct test experiments on FaceForensics++ and Celeb-DF (v2) datasets, and comprehensive results show that ENST has higher accuracy and generalization, which is superior to the most advanced methods.

**Index Terms:** deepfakes, video forgery detection, high frequency, texture, optical flow, EfficientNet, Swin Transformer

## 1 Introduction

Video synthesis techniques based on deep learning Karras et al. (2017; 2018); Thies et al. (2019) have reduced the threshold for people to manipulate videos, and the high-quality deepfake videos that are indistinguishable to the human eye are increasingly available Deepfakes (2020); Faceswap (2019); Fakeapp (2020). Although this technology has been applied to Cineflex, Crytek, and driverless technology, which has had a positive effect on society. Inevitably, it is illegally abused by some malicious users to forge indistinguishable false information, which seriously affects the level of public trust in the government and society, causes irreparable social problems, and can even pose a serious threat to politics.

Many deepfake detection methods Wang et al. (2019); Kim et al. (2021); Jeon et al. (2020) have been proposed to address this problem and significant progress has been made. Previous works explore forgery patterns based on manual features Ferrara et al. (2012); Pan et al. (2012); Cozzolino et al. (2015) and then classify them by zooming in on the subtle differences between the real and forged images. While most of the recent learning-based approaches Cozzolino et al. (2017); Chollet (2017a) define the problem as a binary classification of an image or video with a process of first extracting

the global features of the image using a neural network and then feeding them to a classifier to distinguish between genuine and faked images.

However, with the increasing refinement of deep generative methods, the generated forged videos are becoming increasingly difficult to discriminate, especially some low-quality ones with only a few subtle visual artifacts. Most existing detection methods still focus on specific artifacts, leading to significant challenges in the generalization and effectiveness of detection systems in realistic scenarios. This creates an urgent need to discover subtle cues that are more critical for detection, and then to extract and incorporate appropriate modules with efficient backbone networks to enhance these artifact features and combine them effectively for forgery detection.

After systematic analysis, we find three fundamental reasons for the low generalization of the model. The first is the difficulty of capturing general artifact cues and the limitations of the dataset in terms of quantity and quality. The second is the limitations of selecting a suitable network model for specific feature extraction. The third is the limitations of fully and effectively utilizing the extracted features. In terms of feature cues, we find that although the generated videos are realistic with few artifacts in the texture, the frequency domain, especially at high frequency, still reveals significant differences, even for low-quality videos, and many forgery methods are designed to generate videos that are slightly defective to human vision, with few methods targeting the frequency domain. In addition, it is found that texture information located at the shallow level of the detection network rather than the higher-level semantic information is particularly critical for detection. To eliminate the effect of luminance in RGB on texture feature extraction, we convert the images from RGB to YCbCr and remove the luminance (Y) channel, and combine Cb and Cr channels with high frequencies. In addition, we find a characteristic that the optical flow of the real video changes significantly while the optical flow of the deepfake video has little change, and coupled with the temporal consistency of the continuous frames of the real video, we choose the high frequency, texture, and optical flow as the feature cues for video forgery detection.

For features such as high frequency and texture, which require attention to local details, we choose EfficientNet-B5 Tan & Le (2019), a CNN class network that is superior in local feature extraction, as one of the backbone networks, which has higher accuracy and performance compared with other CNNs. For features such as optical flow, which reflect the inter-frame temporality, we choose Swin Transformer Liu et al. (2021), a Transformer class network that is superior in global representation, as another backbone network, which applies to all scales of images, simple and flexible in computation, and shows excellent performance compared with other Transformers. In addition, we introduce the attention mechanism and a new loss function for EfficientNet-B5 to extract more robust face features.

In summary, in this paper, we propose a novel multiple-cue video forgery detection model ENST using a combination of high frequency, low-level texture, and optical flow cues, based on the fusion of EfficientNet-B5 and Swin Transformer, which fully integrates multiple key features and uses an advanced backbone network for extraction. To demonstrate the effectiveness of ENST, we conduct extensive experiments on FaceForensics++ Rssler et al. (2019) and Celeb-DF (v2) Li et al. (2019), as well as a comparison of various settings in an ablation study. Comprehensive experiments indicate that ENST has excellent competitive properties.

The contributions of this paper are as follows:

**1.** We investigate the current video forgery detection methods in-depth and systematically, and find three reasons for the poor generalization ability of these methods, as well as discover several artifact cues in forgery detection, and use these cues to design an efficient network structure for the above reasons.

**2.** We propose a novel network structure ENST, which can utilize the local feature extraction ability of EfficientNet-B5 and the global relationship sensing ability of Swin Transformer, and integrate them to complement their abilities during data processing. In addition, we introduce the attention module and a new loss function for EfficientNet-B5 to enhance the ability to extract features.

**3.** Our experiments on FaceForensics++ and Celeb-DF (v2) demonstrate that ENST achieves superior classification performance and generalization compared to other state-of-the-art methods.

The rest of this paper is organized as follows. Section 2 presents related work in the field of video forgery detection and describes the motivation for the proposed ENST. Section 3 details the specifics

of the proposed model. Section 4 describes the experimental results and analysis. Section 5 concludes this paper.

## 2 RELATED WORK

### 2.1 VIDEO FAKE CUES

In recent years, various deep learning forgery methods have produced synthetic images or videos of faces Suwajanakorn et al. (2017; 2015) that have caused some degree of harm to society. The problem of deep forgery detection has received extensive attention in the vision field, and many forgery detection methods have been proposed to detect such forged products. Some methods focus on mining specific artifacts generated by the forgery process, such as color space Li et al. (2018); McCloskey & Albright (2018) and shape cues. Many deep learning methods Marra et al. (2018); Yu et al. (2018); Zhou et al. (2018) use deep neural networks to extract high-level semantic information from the spatial domain and then classify a given image or video. Some methods convert the image from the spatial domain to the frequency domain Stuchi et al. (2017); Franzen (2018), capturing some information that is valuable for forgery detection. Stuchi et al. (2017) employ a fixed set of filters to extract different ranges of frequency information and then adopt a fully connected layer to obtain classification results. Durall et al. (2019) extract the frequency domain information using the DFT transform and average the amplitudes of the different frequency bands. Other methods extract statistical features, such as capturing features from spatial textures Pan et al. (2009); Conotter (2011) and transform domain coefficient distributions Chen et al. (2009); Lyu & Farid (2005). Video detection methods can discover differences in the temporal dimension of video frames Sabir et al. (2019); Guera & Delp (2018), such as Wang & Dantcheva (2020) using spatial and motion information, and Chintha et al. (2020); Amerini et al. (2020); Caldelli et al. (2021) using optical flow for video forgery detection. While the above methods are limited to specific cues, we take full advantage of these cues.

### 2.2 DEEPFAKE DETECTION STRUCTURES

Many deep learning models have been proposed to extract features and discover forgery patterns directly without targeting a specific forgery cue or artifact, classify real and fake images or videos, and implement deep forgery detection created by random methods. EfficientNet Tan & Le (2019) achieves higher accuracy with a larger network by combining a composite model expansion method with neural structure search techniques to optimize the dimensions of network depth, network width, and input image resolution, which has higher accuracy and performance compared to other advanced CNNs. Each layer of Swin Transformer computes self-attention within the window of the transform, enabling cross-window connectivity for all scales of the image. Swin Transformer is flexible and efficient, with only linear complexity for image size. Convolution module in CNN is good at extracting local details, and as the layers get deeper, the attention field will be wider and more suitable for image processing. But to capture global information, detection model often requires stacking many convolutional layers. While attention in Transformer is good at grasping the whole, but requires a large amount of data for training. Conformer Peng et al. (2021) uses a parallel and complementary approach, combining both CNN and Transformer branches to preserve the features extracted by both branches, to achieve performance beyond CNN and ViT Dosovitskiy et al. (2020) with comparable parameter complexity.

## 3 PROPOSED METHOD

In this section, we outline the principles and details of the different parts of the proposed method according to the data processing process.

It is challenging for most methods to extract sufficiently comprehensive features with suitable network models and then integrate them effectively for forgery detection. Inspired by Conformer Peng et al. (2021), we choose two models EfficientNet-B5 and Swin Transformer with more efficient performance among CNNs and Transformers, as baselines and combine them for video forgery detection.

EfficientNet-B5 extracts local features of frames, such as high frequency features and texture features by using the characteristic of focusing well on local features. Swin Transformer extracts artifact features from the optical flow map between frames to obtain the global representation of frames, where the optical flow features responding to the temporal consistency between frames are used.

We design a parallel interactive deep forgery detection architecture ENST to learn a potential representation that can distinguish between real and forged faces, and the pipeline of ENST is shown in Figure 1.

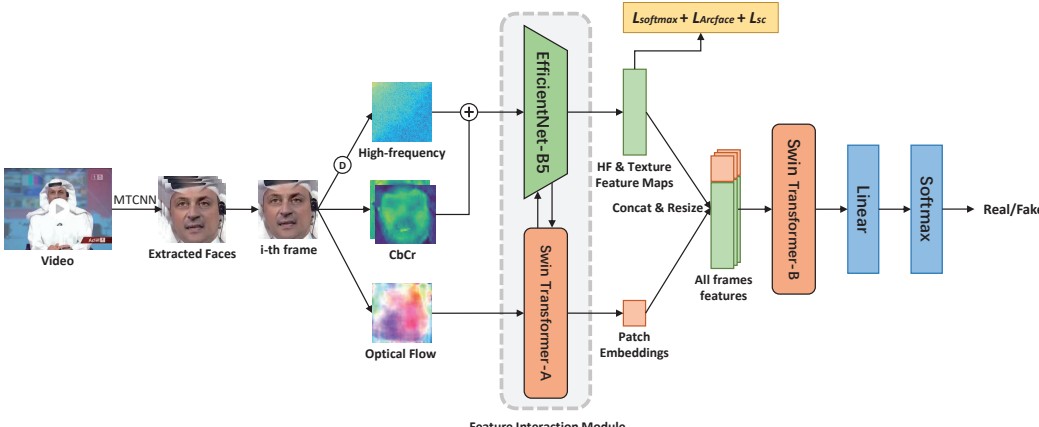

Figure 1: The video forgery detection pipeline of proposed ENST, where Swin Transformer-A represents the first three stages in Swin Transformer, Swin Transformer-B represents the last stage in Swin Transformer, and D represents DCT operation.

## 3.1 PRE-PROCESSING

We first extract the frames of the input video, and then use the Multitask Cascaded CNNs (MTCNN) Zhang et al. (2016) to detect and extract the faces present in each frame, crop the faces of each frame, then resize to 224×224 pixels. Moreover, we normalize them to zero mean and unit variance, and finally obtain the extracted faces. After that, we first operate on a single frame in the video, and the face in frame $i$ is input to two branches after the basic feature extraction process.

## 3.2 EFFICIENTNET-B5 BRANCH

We exploit the structurally improved EfficientNet-B5 branch to extract the deepfake information in the high frequency and texture, and enhance it to finally obtain the features of a fully connected linear layer. We introduce the specific details in the following subsections.

### 3.2.1 FEATURE

The motion process of faces is complex, and most forgery methods do not consider sufficiently precise global constraints during forgery, which can easily lead to missing facial texture details of the generated faces and inconsistencies among features in diverse domains.

As the skin tones in RGB images are more influenced by the luminance, we convert the extracted faces from RGB to YCrCb by separating and removing the Y channel which denotes luminance.

In addition, considering the important effect of high frequency on forgery detection, we convert the image from the spatial domain to the frequency domain by DCT operation, and then use a high-pass filter to extract the high frequency component, which was merged with the two channels of Cr and Cb to form a three-dimensional 224×224×3 feature tensor. Then it was fed into EfficientNet-B5 to extract the detail components in high frequency and the fine artifacts of the shallow texture.

### 3.2.2 Loss

We assume that real and fake faces in videos have distinguishable feature distributions, and samples of different classes are clustered together. To extract robust face features and to distinguish the video distributions of real and fake faces, instead of using the more common softmax and cross-entropy, we combine softmax loss, additive angular margin loss (ArcFace) Deng et al. (2018) and single-center loss (SCL) Li et al. (2021) as the loss function of EfficientNet-B5 for extracting features. ArcFace and SCL have some similarities, both are used to compress intra-class tightness and enhance inter-class diversity. And we combine these objective functions to improve the accuracy of the extracted features.

**ArcFace**

ArcFace is an improvement on SphereFace Liu et al. (2017) in terms of feature vector normalization and additive angular spacing, forcing a boundary between the distance of a sample to its class center and the distance of a sample to other class centers in a corner space. It improves interclass separability, and also enhancing intraclass tightness and interclass diversity, allowing the model to learn features that are highly distinguishable between real and fake faces features, resulting in a more robust classification for forgery detection. ArcFace loss is defined as:

$$L_{Arcface} = -\frac{1}{N}\sum_{i=1}^{N}\log\Big(\frac{e^{s\cdot(\cos(\theta_{y_i}+m))}}{e^{s\cdot(\cos(\theta_{y_i}+m))} + \sum_{j=1, j\neq y_i}^{N} e^{s\cdot\cos\theta_j}}\Big), \tag{1}$$

where $N$ is the batch size, $n$ is the class number, $m$ is an additive angular margin penalty between $x_i$ and $W_{y_i}$, $\theta_j$ is the angle between the weight $W_j$ and the feature $x_i$, $W_j$ is the $j$-th column of the weight $W$, and $s$ is the radius of a hypersphere where the learned embedding features are thus distributed on.

**Single-center Loss (SCL)**

The purpose of SCL is to minimize the distance from the real face to the center point while maximizing the distance from the fake face to the center point, so that the network can learn more subtle forgery information and reduce the optimization difficulty. SCL is defined as:

$$L_{sc} = M_{nat} + \max(M_{nat} - M_{man} + m\sqrt{D}, 0), \tag{2}$$

where $M_{nat}$ is the average Euclidean distance of the true face representation from the center point $C$, and $M_{man}$ is the average Euclidean distance of the false face representation from the center point $C$. The Euclidean distance $m$ is related to the arithmetic square root of the feature dimension $D$. To facilitate the setting of the hyper-parameter $m$, the boundary is designed here as $m\sqrt{D}$.

In addition, considering that SCL is based on mini-batch samples and focuses directly on feature representation, while softmax loss can focus on the global and on how to map the feature representation to the discrete label space. So, we use the global information retained by softmax loss to guide the update of center points in SCL and increase the robustness of training.

In general, to reconcile the local feature representation and the global update, we use a combination of three loss functions, The total loss function is defined as:

$$L_{total} = L_{softmax} + \alpha L_{Arcface} + \beta L_{sc}, \tag{3}$$

where $\alpha$ and $\beta$ are hyper-parameters or weights that adjust the balance between $L_{softmax}$, $L_{Arcface}$, and $L_{sc}$. They are used to provide a relatively efficient and flexible total loss function.

### 3.2.3 Attention

The attention mechanism allows models to focus on and fully learn crucial information, and is widely adopted as a module in models in natural language processing and computer vision Jie et al. (2017); Huang et al.; Vaswani et al. (2017).

We take EfficientNet-B5 as the extraction model for artifact features in high frequency and low layer textures. To focus more on the artifacts in the feature maps, inspired by Bonettini et al. (2020), we insert the attention module between the MBConv layers of EfficientNet-B5, as shown in Figure 2,

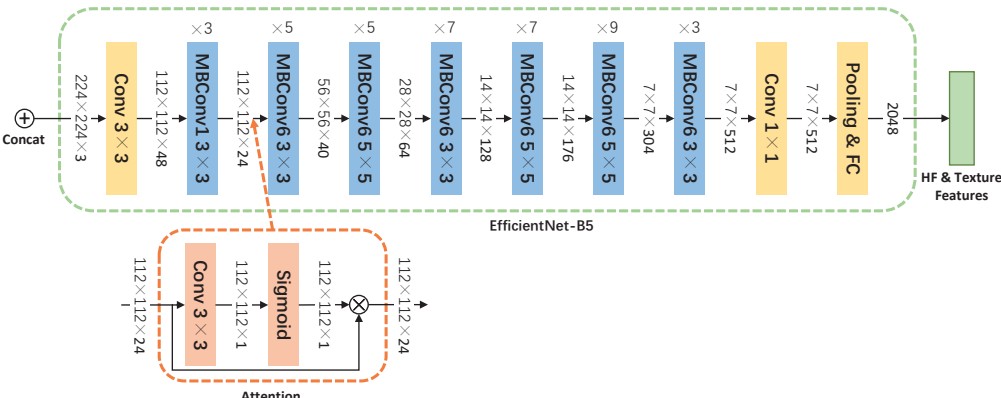

Figure 2: Attention module is introduced into MBConv interlayers in EfficientNet-B5.

Figure 2 shows the effect of adding only one attention module. In the practical experiment, we add the attention module sequentially from front to back between layers to compare the effect of the attention mechanism on the overall detection performance of the model. First, the high frequency and texture information of the frames are combined and fed to EfficientNet-B5. Forging the images will result in overlapping areas of high frequency and texture information, which can be a guide to the extraction of texture information. In addition, the attention module can notice these subtle artifacts in the shallow layers. The final output is an enhanced fusion feature of high frequencies and textures.

### 3.3 SWIN TRANSFORMER BRANCH

Considering that the optical flow of the real video changes, while the optical flow of the faked video rarely changes. In addition, the distance between parts of the face and the head motion vary continuously and regularly between frames of the real video, while most forgery methods seldom concern to ensure the continuity between frames of the deep forgery video by effective modeling, which leads to the temporal inconsistency of the consecutive frames.

To exploit this temporal variation difference, we first divide the video into $0 \sim N$ consecutive frames, and extract the optical flow of frame $i$ and frame $i + 1$ as the optical flow of frame $i$ by using PWC-Net Sun et al. (2018). Then we input them into Swin Transformer-A to obtain the patch embeddings of the middle layer, where Swin Transformer-A represents the first three stages in Swin Transformer.

### 3.4 FEATURE INTERACTION MODULE

We introduce the feature of complementary fusion by adding a feature interaction module. In order to solve the problem of mismatching the size of the feature map in the Efficient-B5 branch and the patch embeddings in the Swin Transformer branch, we use a special conversion operation, in which a 1×1 convolution is used to align the dimensionality of the feature map with the number of channels of the patch embeddings. Then, the spatial dimensions are aligned by the downsampled module. Finally, the feature map are added to the patch embedding.

In addition, the global context is gradually exported from the Swin Transformer branch to the feature map in Efficient-B5 to enhance the global perception capability of the Efficient-B5 branch. We adopt a similar operation by first upsampling the patch embedding to align the spatial scales. Then a 1×1 convolution is used to align the number of channels of the patch embeddings with the dimensionality of the feature map.

Moreover, considering that the forgery operation causes changes in various aspects of the forgery region, the same regions of different features extracted have definite similarities, so this parallel and complementary fusion of the two branches also enables the two branches to guide each other and focus on the local artifact that is instrumental for forgery detection.

### 3.5 FEATURE CONCATENATION

This part is the combination of all face region features of frame $i$ extracted from two branches connected to form a feature concatenation of frame $i$, which includes the extracted high frequency, texture features, and patch embeddings.

The above operations are performed for each frame in sequence to obtain the feature concatenations of the $0 \sim N$ frames of the videos. After resizing, they are transformed into individual patches, which are combined into $N$ patches, and then transformed into a new patch embedding, and then input it to Swin Transformer-B, then followed by the linear layer and softmax layer, finally output the classification prediction of the whole video, where Swin Transformer-B represents the last stage in Swin Transformer. It is worth mentioning that, after "$i$-th frame" and before "All frames features" in Figure 1 represent the operation process of the $i$-th frame alone, and the rest of Figure 1 represents the operation process of all frames.

## 4 EXPERIMENTS

### 4.1 DATASETS

In this paper, we use FaceForensics++ to train and verify the performance of our proposed model ENST, and Celeb-DF (v2) to verify the generalization of the model.

#### 4.1.1 FACEFORENSICS++

FaceForensics++ is a facial forgery video dataset, including original videos and videos generated by the four most representative face manipulation methods (Face2Face, FaceSwap, DeepFakes and NeuralTextures), each category has 1000 forgery videos, and the extracted frames contain more than 500,000 forged images in each category. According to different compression levels, the compression factors can be classified as c0(raw), c23(medium) and c40(high). As with the setup in FaceForensics++, we split the dataset in each category into a fixed train set, a validation set and a test set containing 720, 140 and 140 videos, respectively. In the following experiments, we use two compression levels of c23(medium) and c40(high) for the data. By default, the metric are Acc and AUC.

#### 4.1.2 CELEB-DF (V2)

Celeb-DF (v2) contains both real and fake videos from Youtube. Unlike the FaceForensics++, the fake videos in Celeb-DF (v2) are further segmented to handle issues such as color inconsistency and low-frequency smoothing, and have online playback-level high-resolution pictures quality. Celeb-DF (v1) contains 795 fake videos. Celeb-DF (v2) is an extension of Celeb-DF (v1) and includes 590 original videos collected from YouTube with different age, race, and gender themes, and 5639 corresponding fake videos. Celeb-DF (v2) has more diverse and challenging data, which is beneficial to verify the generalization of the proposed method.

### 4.2 SETTINGS

Based on the parameters of Conformer by default, we combine the parameters in both branches of EfficientNet-B5 and Swin Transformer. ENST uses an optimizer of AdamW. The momentum is 1.0. The weight decay factor is 0.05. The batch size is 256. The epoch is 300. The initial learning rate is 0.001. The LR scheduler is decaying with a cosine schedule.

### 4.3 COMPARISON OF METHODS

We compare the proposed method ENST with previous high-performance video forgery detection models on FaceForensics++, and the results are shown in Table 1. Comparing with some advanced methods, we find that although c40 reduces the AUC by 6.50 and the accuracy by 10.57 compared to c23, it still outperforms other methods. We analyze that although the compression loses the details of image texture, the extracted high-frequency information and the optical flow information between

Table 1: Comparison of the results of different models on the two quality types of FaceForensics++.

| Model | c40 | | c23 | |
|---|---|---|---|---|
| | AUC | Acc | AUC | Acc |
| MesoNet Afchar et al. (2018) | - | 70.47 | - | 83.10 |
| Face X-ray Li et al. (2020) | 61.60 | - | 87.40 | - |
| Xception Chollet (2017b) | 89.30 | 86.86 | 96.30 | 95.73 |
| Two-branch Masi et al. (2020) | 86.59 | - | 98.70 | - |
| Efficient-B4 Zhao et al. (2021) | 90.40 | 88.69 | 97.60 | 99.29 |
| **ENST(Ours)** | **91.12** | **88.73** | **97.62** | **99.30** |

video frames have excellent robustness to the compressed information, which can well resist the compression interference and maintain the robustness of the model.

Table 2: Comparison of generalization results of different models, which train on the FaceForensics++ train set and test on Celeb-DF (v2) (AUC(%)).

| Model | FF++ | Celeb-DF (v2) |
|---|---|---|
| Two-stream Zhou et al. (2018) | 70.10 | 53.80 |
| MesoInception4 Afchar et al. (2018) | 83.00 | 53.60 |
| Two Branch Masi et al. (2020) | 93.18 | 73.41 |
| Efficient-B4 Zhao et al. (2021) | 99.8 | 67.44 |
| **ENST(Ours)** | 99.8 | **67.51** |

In order to verify whether ENST has superior generalization in realistic scenarios that are not included in the train set, we put the model trained on FaceForensics++ on Celeb-DF (v2) for experiments on transfer learning, we use the fine-tuning model, i.e., freeze parts of the convolutional layers of the pre-trained model, train the remaining convolutional layers, and after the model converges, reduce its learning rate and fine-tune the layers. We randomly select 40 samples in Celeb-DF (v2) with FaceForensics++ to train the model to prevent catastrophic forgetting, and the results are shown in Table 2. The experimental results show that ENST has a better transfer capability and the ability to learn new types of samples. It shows superior generalization on new types of data.

## 4.4 ABLATION STUDY

To verify the effectiveness of each part of ENST, we conduct an ablation study and evaluate the network on FaceForensics++.

### 4.4.1 COMPARISON OF LOSS

Table 3: Comparison of results using different loss functions in EfficientNet-B5 on c23 of the FaceForensics++.

| Loss Function | AUC | Acc |
|---|---|---|
| Arcface | 88.15 | 84.48 |
| SCL | 92.27 | 90.64 |
| Softmax + Arcface | 91.52 | 87.86 |
| Softmax + SCL | 95.46 | 96.93 |
| **Softmax+ Arcface + SCL** | **98.67** | **97.65** |

On FaceForensics++, we compare the experimental results of using different loss functions when extracting features in EfficientNet-B5, as shown in Table 3. First, we compare the classification results of Arcface and SCL individually, after which softmax is added to guide the improvement of the global classification. Finally the three losses are combined. The experimental results indicate that the highest accuracy is achieved when softmax, Arcface, and SCL are integratively used.

Table 4: Comparison of results on the FaceForensics++ with the addition of the attention module at different stages of EfficientNet-B5, where ✓indicates the addition of the attention module in stage i of EfficientNet-B5. Stage 1 indicates after the first MBConv. Only the first five of the seven are listed here.

| | | | | | | | |
|---|---|---|---|---|---|---|---|
| | 1 | | ✓ | ✓ | ✓ | ✓ | ✓ |
| | 2 | | | ✓ | ✓ | ✓ | ✓ |
| **Stage** | 3 | | | | ✓ | ✓ | ✓ |
| | 4 | | | | | ✓ | ✓ |
| | 5 | | | | | | ✓ |
| **AUC** | | 96.35 | 97.82 | 98.53 | **98.67** | 98.55 | 95.63 |
| **Acc** | | 94.65 | 96.07 | 96.53 | **97.65** | 94.49 | 89.48 |

### 4.4.2 COMPARSION OF ATTENTION

On FaceForensics++, we start from the previous of the EfficientNet-B5 structure and add attention modules at different stages, and the experimental results are shown in Table 4, which shows that the accuracy increases gradually with the addition of attention modules. However, when continuing to add at the 4th stage, the accuracy has slightly decreased, and when continuing to join in the 5th stage, the accuracy rate shows a significant decrease, which is consistent with our speculation. Our analysis suggests that the artifacts are mainly concentrated in the shallow network of EfficientNet-B5, and the shallow feature extraction can provide enough critical information for the whole classification network to classify, while the deep feature extraction is of little significance. Furthermore, the increase of the attention module will increase the computational volume and affect the overall performance improvement.

## 5 CONCLUSION

In this paper, we systematically analyze the reasons for the low generalization of the current forgery video detection system and discover some subtle but critical cue features in forgery detection. In order to combine the discovered cues and improve the generalization problem, we propose a network structure ENST with parallel interaction fusion of two branches, EfficientNet-B5 and Swin Transformer. We also introduce the attention mechanism and add a new loss function to EfficientNet-B5 to further focus on and enhance the artifact features and extract more robust face features. Comprehensive experiments indicate that ENST has better performance and generalization than other advanced methods.

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
