# OpenReview forum: "Video Forgery Detection Using Multiple Cues on Fusion of EfficientNet and Swin Transformer"
_ICLR.cc/2022/Conference — ICLR 2022 Submitted_

### Official Review · Reviewer_8erJ · 2021-10-29

**Correctness:** 4
**Technical Novelty And Significance:** 3
**Empirical Novelty And Significance:** 3
**Recommendation:** 8
**Confidence:** 4

**Main Review:**

Pros:
1. This paper is well structured and very easy to follow. The authors analyzed why generalization is hard for video forgery detection and give a clear motivation/description of proposed parallel network.
2. The idea of fusing multiple artificial clues and introducing interactive between two branches to guide each other is novel and makes sense to me.
3. The proposed new loss function and attention mechanism and the ablation study for them make the proposed method more convincing.

Cons:
I think the experiment part can be further improved.
1. The compared methods in Table1 and Table2 are not the same, which is confusing. To make the comparison more clear, I think the authors can give one sentence summary for each of methods just to give readers a sense of the difference between proposed method and compared method.
2. The authors claim that proposed method has better generalization but the experiment conducted for Table2 is not convincing to me.
    1. Details of transfer learning/fine-tuning are missing. What’s effect of fine-tuning process on generalization?
    2. The numbers reported in Table2 is AUC or Acc?
3. I like the idea of two-branch network structure, but to back up this design, it will be stronger to conduct ablation study to investigate the contribution of each branch and the effect of feature interaction between two branches.

**Summary Of The Paper:**

This paper proposes a new video forgery detection method (ENST) by combining multiple forgery clues including high frequency, low-level texture, and optical flow. ENST employs a two-branch network, an EfficientNet-B5 branch for high frequent and texture info and an Swin transformer branch for optical flow info. The motivation behind this design is that forgery traces are more likely to present in texture regions where human eyes cannot easily catch, and the optical flow of forgery videos has rare variations compared to real videos.

**Summary Of The Review:**

I would recommend accepting this paper. The technique part looks sound to me. The two-branch network structure, the feature interaction between EfficientNet-B5 and Swin transformer, as well as the use of attention and new design of loss function for EfficientNet-B5 make the proposed method solid and convincing. My major concern is from experimental results and ablation study.

---

> ### Author Response · Authors · 2021-11-10
> **Thanks for your understanding and recognition. Your careful and valuable comments are very important to me. We will explain your concerns point by point.**
>
> Answer1: Yes, we will supplement it in the revised version later.
>
> Answer2.1: In transfer learning, catastrophic forgetting phenomenon often occurs, that is, the test effect in another domain will be greatly reduced. One method is to add some data in the training data set to be transferred for training, which can alleviate this phenomenon. So you get better generalization.
>
> Answer2.2: The numbers in Table 2 refer to AUC, now it has been added.
>
> Answer3: Your suggestion on the ablation experiment of two-branch network structure is very valuable, and we will adopt it in the revised version later.
>
> If you still have any questions, welcome to discuss again, thank you.

---

### Official Review · Reviewer_eCnG · 2021-11-02

**Correctness:** 2
**Technical Novelty And Significance:** 1
**Empirical Novelty And Significance:** 2
**Recommendation:** 3
**Confidence:** 5

**Main Review:**

My main concern about this submission is related to the contribution, that in my opinion is limited with respect to the current literature. In fact, many of the observations/findings made by the authors are well known in the forensics field and already exploited in published papers. More specifically:

- The importance of high-frequency features has been exploited in many papers that often work on the so called noise residuals obtained suppressing the scene content, see basic works on constrained CNNs (Cozzolino et al. 2017, Bayar et al. 2018). Also frequency domain analysis has been applied to better enhance such frequency domain features (Frank et al. 2020).

- Temporal inconsistencies have been largely exploited especially for deepfake detection, using different types of analyses, even considering optical flow like in this work, see (Amerini et al. 2020, Caldelli et al. 2021).

- Attention mechanisms have been included in several solutions, also similar to what has been done in this work, see (Dang et al. 2020, Bonettini et al. 2020)

- Dual-branch structures that also include a fusion step have been used in several papers, see (Zhou et al. 2018, Masi et al. 2020).

- Working in the YCbCr domain has been done in (Li et al. 2018).

It is also worth observing that EfficientNet-B7 has been used by the winner of the Kaggle competition in 2020, while transformers have been already considered for deepfake detection in some recent publications (Coccomini et al. 2021, Zheng et al. 2021).

For the experimental analysis one can observe a very limited improvement in terms of AUC and Accuracy with respect to Zhao et al. 2021, see Table 1 and Table 2. Note that in Zheng et al. results on Celeb-DF are higher than those reported in Table 2 of this work (do these numbers refer to AUC or Accuracy?). Maybe it is worth to check this misalignment (see Table 4 in Zheng et al.). In addition, for what I can understand the experiment carried out in Table 2 uses some videos of Celeb-DF for fine-tuning. I think that if you want to show generalization, you should not include those videos in training.

Generalization is tested only considering two datasets, FaceForensics++ and Celeb-DF. In my opinion a more extensive analysis should be carried out (see for example the Kaggle dataset proposed by Facebook) and more competitive methods should be included for comparison.

Minor comment: lack of information for some references in the bibliography (venue of publication is missing) and many typos
(e.g. Rrf Attux, Niener, Rssler). The reference by Zhou et al. is repeated twice.

References
- Bayar et al. Constrained Convolutional Neural Networks: A New Approach Towards General Purpose Image Manipulation Detection, IEEE TIFS 2018
- Dang et al. On the detection of digital face manipulation, CVPR 2020
- Frank et al. Leveraging Frequency Analysis for Deep Fake Image Recognition, ICML 2020
- Coccomini et al. Combining EfficientNet and Vision Transformers for video deepfake detection, arXiv 2021
- Zheng et al. Exploring Temporal Coherence for More General Video Face Forgery Detection, CVPR 2021

**Summary Of The Paper:**

This paper faces the problem of video deepfake detection. The idea is to rely on high frequency, texture and optical flow cues to gain generalization. To this end it is proposed a dual-branch detection approach: low-level features are extracted by EfficientNet-B5 that also includes an attention mechanism, while temporal inconsistencies are captured by Swin Transformer. Experiments are carried out on FaceForensics++ and Celeb-DF (v2) and show better generalization ability with respect to state-of-the-art.


**Summary Of The Review:**

In this paper it is proposed a deep learning based approach for video deepfake detection. In my opinion the technical novelty is not sufficient to warrant publication in ICLR and the performance improvement is marginal with respect to state-of-the-art.

---

> ### Author Response · Authors · 2021-11-10
> **Thanks for your careful and valuable comments. We will explain your concerns point by point.**
>
> Contribution:
> In my opinion, the biggest contribution of this paper lies in the effective combination of known clues, structures and loss functions. As far as I know, existing methods don't want to do that, but it is very effective and can significantly improve generalization and robustness, which I believe has also been confirmed in the experiment.
>
> Experimental analysis:
> Firstly, the numbers in Table 2 refer to AUC (modified). Having high classification results is part of the goal of this paper. The previous work has already had high classification results, and it is difficult to improve them greatly on the Internet. In addition, this paper mainly focuses on generalization.
>
> Generalization:
> It is hoped that there will be more experiments to supplement and enhance.
>
> References:
> Yes, there are some problems in the reference, now it has been revised, and I hope it can be resubmitted.
>
> If you still have any questions, welcome to discuss again, thank you.

---

> > ### Comment · Reviewer_eCnG · 2021-11-27
> > **Thanks for your answers**
> >
> > First, I want to thank the authors for their answers to my comments.
> > In my opinion the effective combination of known ideas is not enough to warrant a new publication,
> > especially if the experimental analysis is limited as it happens in this work.
> > As already said in my comments, if the aim is to show generalization then videos from the data under test should not be used to fine-tune the model.
> >
> > Overall, I believe this submission requires more work in order to be considered acceptable at ICLR and I will keep my previous rating.

---

### Official Review · Reviewer_3xwe · 2021-11-02

**Correctness:** 3
**Technical Novelty And Significance:** 2
**Empirical Novelty And Significance:** 2
**Recommendation:** 3
**Confidence:** 5

**Main Review:**

### Strengths:
1) Important topic
2) Identifies important issues in previous deepfake detectors.

### Weakness:
1) The weakness that author mentions are the main contribution of this paper are already identified by [a].
2) The paper lacks novelty as the proposed method is using previously available methods and new additions are not significant.
3) There are many places in the paper, where the author makes some claims but they are not supported by any reference.
4) The paper is not well written and need rewriting in many sections. For example the use of word “inability” in page 2 paragraph 3 is not correct.
5) The literature reviews lacks many recent methods [a], [b], [c], [d] and many more. I will suggest the author to look at these papers are find more related papers from them.
6) The empirical analysis lacks comparison with recent SOTA methods such as CLRNet [a], FReTAL [b], TAR [c], CoReD [d] etc.

[a] Tariq, S., Lee, S., & Woo, S. (2021, April). One detector to rule them all: Towards a general deepfake attack detection framework. In Proceedings of the Web Conference 2021 (pp. 3625-3637).

[b] Kim, Minha, Shahroz Tariq, and Simon S. Woo. "FReTAL: Generalizing Deepfake Detection using Knowledge Distillation and Representation Learning." Proceedings of the IEEE/CVF Conference on Computer Vision and Pattern Recognition. 2021.

[c] Lee, Sangyup, et al. "TAR: Generalized Forensic Framework to Detect Deepfakes Using Weakly Supervised Learning." IFIP International Conference on ICT Systems Security and Privacy Protection. Springer, Cham, 2021.

[d] Kim, M., Tariq, S., & Woo, S. S. (2021, October). Cored: Generalizing fake media detection with continual representation using distillation. In Proceedings of the 29th ACM International Conference on Multimedia (pp. 337-346).







**Summary Of The Paper:**

The author proposes a method for detecting deepfakes that makes use of EfficientNet B5 and Swin Transformer. Additionally, the authors propose a loss function and an attention mechanism for EfficientNet B5. The proposed methods demonstrate increased accuracy and generalizability when compared to the baselines.

**Summary Of The Review:**

The paper lacks novelty and writing need improvement.

---

> ### Author Response · Authors · 2021-11-10
> **Thanks for your careful and valuable comments. We will explain your concerns point by point.**
>
> Answer1&2: Our main innovation or goal is not the novelty of the clue, but to be able to combine these models and clues in an effectively way to maximize their advantages, rather than a separate forgery detection to improve the generalization of the unseen sample, and the implementation proves the effectiveness of this combination. As far as we know, the most fundamental reason why the current method is weak in generalization is that it does not combine cues and loss functions from multiple perspectives, and I think this paper is different from other papers.
>
> Answer3&5: I hope to revise this in future releases.
>
> Answer4: Yes, overall, I think the article is easy to understand, and we will also review the writing of the article.
>
> Answer6: I have looked at the article you mentioned, and the model proposed in this paper also has great advantages in classification compared with it. In addition, the main purpose of this paper is to provide a way to improve generalization, rather than SOTA.
>
> If you still have any questions, welcome to discuss again, thank you.

---

> > ### Comment · Reviewer_3xwe · 2021-11-22
> > **Thank you for the response**
> >
> > I've taken the time to read the author's response. I am still not convinced that this paper contains sufficient technical novelty to warrant acceptance to ICLR. As a result, I will adhere to my initial decision.
> >
> > Additionally, I encourage the author to also conduct future research on audio-video deepfakes.

---

### Official Review · Reviewer_c2vw · 2021-11-05

**Correctness:** 3
**Technical Novelty And Significance:** 2
**Empirical Novelty And Significance:** 2
**Recommendation:** 3
**Confidence:** 4

**Main Review:**

Strengths
1. The proposed model integrates the information of frequency domain, spatial domain and optical flow to detect video forgeries.
2. The paper is well written and easy to follow.

Weaknesses
1. The proposed multi-branch framework has already been used in previous works; the information of frequency domain, spatial domain and optical flow used in video/image forgery detection have been explored in previous works; the loss functions, i.e., ArcFace and SCL, and the attention module are all borrowed from existing methods. The novelty of this paper seems limited.
2. This proposed framework utilizes two of the most advanced model structures, i.e., EfficientNet and Swin Transformer, but the results in Tables 1 and 2 only show very marginal improvement.
3. The authors claimed three clues that are helpful for the video forgery detection, i.e., 1, high-frequency information, 2, texture in the shallow layer of the model, 3, the optical flow of the real video has variations while the optical flow of the deepfake video has rarely variations. While the first has already been explored while the second one is not surprised as well, it would be interesting to see the evidence (examples) of the clues, especially for the third one.




**Summary Of The Paper:**

The proposed multi-branch framework that integrates the information of frequency domain, spatial domain and optical flow to detect video forgeries. It also applies two of the most advanced model structures, i.e., EfficientNet and Swin Transformer.

**Summary Of The Review:**

My majorpr concern is the novelty.

---

> ### Author Response · Authors · 2021-11-10
> **Thanks for your careful and valuable comments. We will explain your concerns point by point.**
>
> Answer1: Our main innovation is not the novelty to use a clue, but can combine these models and clues ably, to maximize their advantages, to improve the generalization of unknown samples. and the implementation is also confirmed the effectiveness of the way of the combination. Because as far as we know, the current method is weak generalization, a fundamental point is that no combination of clue and the loss function from multi-angle.
>
> Answer2: Yes, we're also aware of this, we think the reason is the high computational complexity of the model make the performance decrease slightly, so the growth is not obvious, should consider to use a more lightweight structure, but the idea we have put forward that the different clues according to the different structure is correct, in addition, the previous method has more obvious advantage on the scores, can have a little ascension has is not easy. In the end, we will reconsider the influence of adopting such a model with a large amount of calculation on the overall performance in limitation part. We believe that replacing it with a similar and lighter backbone will have more obvious effect, but it doesn't take away from the core contribution of the article.
>
> Answer3: You are right. We will consider adding evidence of this clue to show the difference between real and fake video on optical flow diagram.
>
> If you still have any questions, welcome to discuss again, thank you.

---

> > ### Comment · Reviewer_c2vw · 2021-11-29
> > **Thanks for your responses.**
> >
> > I’ve carefully read the authors’ responses and would appreciate their effort.
> > While there are several interesting points in this paper, I think the main contribution that the authors claimed in the rebuttal, i.e., combining multiple clues, has been explored in previous works, e.g., [A].
> > Moreover, the authors claimed in “Answer2” in the rebuttal that the computational complicated models (such as EfficientNet and Swin Transformer) may harm the performance and they would like to try a “more lightweight structure” to improve the performance. While it is a little counterintuitive to me, I think it’s very interesting to have such things in future work.
> >
> > I will keep my previous rating.
> >
> > [A] Y. Qian, et. al. “Thinking in frequency: Face forgery detection by mining frequency-aware clues.” ECCV 2020.

---

### Decision · Program_Chairs · 2022-01-20

**Decision:**

Reject

**Comment:**

The authors propose a new method for deepfake detection (ENST) which relies on high-frequency information, low-level/shallow features, and optical flow. In particular, EfficientNet-B5 is used to extract the high frequency info and shallow features, and a Swin Transformer to capture discrepancies between optical flows. Empirical validation on FaceForensics++ and Celeb-DF shows some improvements over the baselines.

The reviewers found this to be a relevant and timely topic. The reviewers also found that integrating information from the frequency domain, the spatial domain, and optical flow is a promising approach. There were three reviewers suggesting rejection, and one suggesting acceptance. After the rebuttal and discussion phase, the following remaining issues were highlighted:
- **Limited technical novelty** (nearly all components used in this work were already expired in other work).
- Underwhelming empirical improvements given the fact that the model uses EfficientNet-B5 and the SwinTransformer.
- Many claims are still not supported by empirical evidence. For instance, to claim generalisation, an extensive analysis, including more datasets as well as competing methods should be carried out.